# Supplementation with NAD^+^ and Its Precursors to Prevent Cognitive Decline across Disease Contexts

**DOI:** 10.3390/nu14153231

**Published:** 2022-08-07

**Authors:** Jared M. Campbell

**Affiliations:** Graduate School of Biomedical Engineering, University of New South Wales Sydney, Sydney 2052, Australia; j.campbell@unsw.edu.au

**Keywords:** cognitive health, cognitive decline, neuroprotection, NAD, NADH, nicotinamide riboside, nicotinamide mono nucleotide, nicotinamide, dementia, stroke, diabetes, traumatic brain injury

## Abstract

The preservation of cognitive ability by increasing nicotinamide adenine dinucleotide (NAD^+^) levels through supplementation with NAD^+^ precursors has been identified as a promising treatment strategy for a number of conditions; principally, age-related cognitive decline (including Alzheimer’s disease and vascular dementia), but also diabetes, stroke, and traumatic brain injury. Candidate factors have included NAD^+^ itself, its reduced form NADH, nicotinamide (NAM), nicotinamide mononucleotide (NMN), nicotinamide riboside (NR), and niacin (or nicotinic acid). This review summarises the research findings for each source of cognitive impairment for which NAD^+^ precursor supplementation has been investigated as a therapy. The findings are mostly positive but have been made primarily in animal models, with some reports of null or adverse effects. Given the increasing popularity and availability of these factors as nutritional supplements, further properly controlled clinical research is needed to provide definitive answers regarding this strategy’s likely impact on human cognitive health when used to address different sources of impairment.

## 1. Introduction

Cognitive decline can be induced by numerous pathological conditions and events. The most well-recognised form is age-related cognitive decline (i.e., Alzheimer’s disease and other forms of dementia); however, cognitive impairment resulting from acute events such as stroke or trauma can also have long-term impacts on cognitive function, as can long-term exposures such as diabetes and chemotherapy. Regardless of the cause, there are few options for preventing cognitive decline in the face of neurodegenerative conditions. However, the fact that this decline is associated with metabolic dysregulation [1,2]—in particular, lowered levels of the essential redox and enzyme cofactor nicotinamide adenine dinucleotide (NAD^+^) [3]—has attracted significant research attention to the hypothesis that supporting NAD^+^ levels through supplementation with its reduced form NADH or NAD^+^ precursors (e.g., nicotinamide riboside (NR), nicotinamide mononucleotide (NMN), nicotinamide (NAM), and niacin (or nicotinic acid)) could be an effective neuroprotective therapy.

NAD^+^ is produced de novo from tryptophan, primarily in the liver, which also metabolises tryptophan to NAM and then releases it into the serum [4]. The NAM is then taken up by other cells for conversion to NAD^+^ in the NAM salvage pathway [4], which can similarly process the other NAD^+^ precursors: NMN, NR, and niacin. NAD^+^-consuming enzymes produce NAM as a byproduct; this is then processed by nicotinamide phosphoribosyltransferase (NAMPT) to yield NMN, which is transformed back into NAD^+^ by the salvage pathway [4].

NAD^+^ is fundamental to cellular energy metabolism, as it is necessary for the electron exchange from the tricarboxylic acid (TCA) cycle through the electron transport chain (ETC) for the generation of ATP by oxidative phosphorylation. Additionally, it is a key substrate for ADP-ribosylation by poly-ADP-ribose polymerase (PARP) enzymes, protein deacetylation by sirtuins, and cyclic ADP-ribose (cADPR) production by CD38 and CD157 [5]. Numerous mechanisms have been proposed to explain how supplementation in support of NAD^+^ levels can achieve neuroprotection, with evidence from different contexts supporting reduced oxidative stress [6], lowered protein aggregation [7], anti-inflammatory effects [8], the stimulation of neuroprotective HCA2 macrophages [9], the blocking of autophagy [10], and the amelioration of mitochondrial damage and dysfunction [11]. This evidence leads to the conclusion that the relevant pathway is complex and likely multifactorial.

This work reviews the evidence on supplementation and treatment with NAD^+^ precursors for the preservation of cognitive function in all health contexts where it has been studied (research strategy detailed in Figure 1). The main focus is on the primary measures of cognitive function and behaviour. The promising avenues and knowledge gaps are highlighted to help guide future research and translation.

## 2. Dementia

By far the greatest research focus in this area has been applied to the application of NAD^+^ precursors to prevent or ameliorate dementia. According to the volume of research, the main priority has been cognitive decline resulting from Alzheimer’s disease, followed by vascular dementia and age-related cognitive decline with no specified aetiology.

### 2.1. Alzheimer’s Disease

Alzheimer’s disease has a complex, multifactorial aetiology [12], with most research focusing on the accumulation of abnormal, highly insoluble, densely packed filamentary protein structures: extracellular amyloid-β plaques and intracellular tau tangles [13]. The disease also has strong metabolic features, with both central nervous system (CNS)-specific and systemic alterations [14]. As the disease progresses, neurons are driven into a disease state and death, resulting in synapse loss and decreased memory and cognition. Given the high metabolic demand of neurons and the age-related decline in NAD^+^, it has long been suspected that metabolic disruption could contribute to Alzheimer’s disease. Research in mouse models of Alzheimer’s disease has shown that NAD^+^ metabolism is significantly disrupted [12,15].

Most of the work on the impact of supplementation with NAD^+^ precursors has been carried out in animal models. A recent systematic review of NAD^+^, its derivatives, and Alzheimer’s disease in rodent models identified eleven studies that had been published up to December 2020, with the synthesis of their findings supporting the conclusion that treatment with NAD^+^ precursors did restore its levels in the brain, with consequent improvements in learning and memory [16]. The results of the included studies supported the involvement of several mechanisms, including reduced oxidative stress, inflammation, and apoptosis, as well as improved mitochondrial function. Several additional studies with relevant findings were not included in the 2021 systematic review [16], including that of Hou et al. (2021), who showed that treating APP/PS1-mutant mice with the NAD^+^ precursor NR (12 nM in drinking water) for 5 months improved learning and memory [15], and that of Rehman et al. (2021), who induced neuronal dysfunction in an Alzheimer’s disease model via the intracerebroventricular injection of amyloid beta Aβ_1–42_ and showed that NAM treatments (250 mg/kg for 1 week) prevented memory deficits [17]. Additionally, Yao et al. (2017) showed that NMN (subcutaneous injection of 100 mg/kg every other day for 28 days) greatly ameliorated cognitive impairment in a transgenic mouse model of Alzheimer’s disease (APPswe/PS1dE9).

Additionally, several of the very few human studies on the impact of NAD^+^ precursor supplementation on cognitive health were carried out to address its potential role in the treatment of Alzheimer’s disease. Unfortunately, the findings here were more equivocal. One study published in 1996 by Birkmayer found that the treatment of 17 Alzheimer’s disease patients with open-label 10 mg/day NADH disodium salt resulted in improved cognitive function based on the mini-mental state examination (MMSE) and global deterioration scale (GDS) over a treatment period of 8–12 weeks without side effects or adverse events [18]. However, in 2000, Rainer et al. published an attempted duplication with the same treatment applied to 25 dementia patients (including those with Alzheimer’s disease and vascular and frontotemporal dementia) and found no effect on MMSE and GDS scores [19]. The cause of the difference between the findings of the two studies is unclear. The most obvious divergence in their designs was the inclusion of non-Alzheimer’s dementia in the latter study. However, although the actual breakdown was not reported, as Alzheimer’s disease is the most common cause of dementia, the majority of participants were likely affected by Alzheimer’s disease in any case. Despite these papers being published over twenty years ago, no further studies could be found following this line of research.

### 2.2. Vascular Dementia

Vascular dementia results from decreased blood flow to the brain, causing the destruction of tissue, and is the second most common form of dementia after Alzheimer’s disease [20]. As well as the positive effects on cell survival that have been observed for NAD^+^ precursors [21], their potential to relieve vascular dysfunction [22] has also raised the hypothesis that they could be an effective treatment for preventing or ameliorating vascular dementia.

In a rat model of chronic cerebral hypoperfusion (a major cause of vascular dementia), direct supplementation with NAD^+^ (intraperitoneal injection of 250 mg/kg/day for 8 weeks) significantly ameliorated the impairment of learning and memory [11]. In another study, the cognitive impairment caused by chronic cerebral hypoperfusion was shown to be improved in mice by treatment with NAM (intraperitoneal injection of 200 mg/kg/day for 30 days), with significant benefits to learning, memory, anxiety, and depression-like behaviours [23]. Yang et al. (2004) also investigated NAM and showed that its early administration (intraperitoneal injection 500 mg/kg) 2 h after the injection of 1-Methyl-4-phenyl-l, 2, 3, 6-tetrahydropyridine, used here as a model of vascular dementia (although it is more typically applied to model Parkinson’s disease), could decrease error numbers, lessen stimulation time, and prolong residence duration on the safety platform in the step-down test [24]. Delayed administration beyond 2 h resulted in decreased effects. This raises a challenge for the findings of the models cited above—the onset of vascular dementia is not generally immediately recognised, and any treatment received in response to the recognition of its symptoms will inevitably be delayed. As such, the immediate initiation of NAD^+^ precursor treatments in direct response to the instigating exposure is not clinically feasible and, if treatment effectiveness decreases by the hour, the success of clinical translation becomes dubious.

More promising, however, are the collective findings made in a series of related papers on the impact of NMN (intraperitoneal injection of 500 mg/kg/day for two weeks) on the cognitive health of aged mice [25,26,27,28]. The authors found that it significantly improved higher brain function in older mice (24 months) compared to the decreases that were observed relative to young mice (2 months). These studies also found that NMN had significant cerebrovascular protective effects, and even that improved gait was associated with the rescue of neurovascular coupling in principal component analysis (PCA), suggesting that, although this was a model of general ageing rather than induced vascular dementia, these findings should be viewed in the latter context. As such, these results indicate that ongoing supplementation with NAD^+^ precursors could be effective for repairing the cognitive effects of vascular dementia, even when not initiated at the onset of decline.

Another study modelled cerebral small vessel disease by installing a continuous pump of angiotensin II to raise blood pressure and cause hypertension [29]. Mice were then fed NR (300 mg/g body weight) in their food for 28 days. The induced hypertension resulted in decreased short-term memory function, which was rescued by the NR treatment, supporting the conclusion that NAD^+^ precursor supplementation can be effective against multiple aetiologies of vascular dementia.

### 2.3. Age-Related Cognitive Decline

The link between age and cognitive decline is one of the most firmly established phenomena of medicine; however, diagnosing specific causes (i.e., Alzheimer’s, vascular) can be difficult, particularly during the early stages of the disease when interventions would make the most difference. As such, treatments that are effective for general age-related cognitive decline without specificity to an exact cause of dementia are desirable. As discussed above, an investigation of old compared to young mice showed that NMN could improve higher brain function [25,26,27,28]. Although other findings from the work indicated that a major part of its impact was specific to vascular dysfunction, the model itself was of cognitive decline resulting from natural ageing; no specific pathology was exogenously induced. These findings were supported by Hosseini et al. (2019) who showed that the treatment of aged (24 months) rats with NMN (intraperitoneal injection, 100 mg/kg every other day for 28 days) alleviated age-related cognitive impairment, including working, recognition, and reference memories as well as cognitive flexibility [30]. Additionally, Johnson et al. (2018) treated 20-month-old mice with NMN (oral gavage, 300 mg/kg/day for 3 weeks) and found that it ameliorated the cognitive hypersensitivity that they observed in older mice [31].

Other NAD^+^ precursors that have been investigated for their impact on age-related cognitive impairment include NR (2.5 g/kg in food for 3 months), which improved short-term spatial memory in 14-month-old wildtype mice [32], and NADH (intraperitoneal injection, 10–100 mg/kg/day for 10 days), which improved navigation and spatial accuracy in aged (22 months) rats [33].

Overall, preclinical research suggests that NAD^+^ precursors can be an effective intervention for different forms of dementia, warranting careful study in controlled clinical trials. However, still lacking are head-to-head comparisons of the available precursors (NR, NMN, NAM, and niacin), which could help to inform the optimal factor for translation. Furthermore, work is needed to investigate the safety and effectiveness of prophylactic supplementation with NAD^+^ precursors to prevent the onset of age-related cognitive decline—regardless of aetiology—in pre-symptomatic, at-risk people.

## 3. Diabetes

As a chronic disease, the incidence and severity of diabetes generally increases with age, and the accumulating damage to organs and tissues it induces has been linked to cognitive decline and an increased likelihood of dementia. Two studies investigated whether supplementation with NAD^+^ precursors could mitigate this effect. Chandrasekaran et al. (2020) administered NMN (subcutaneous injection, 100 mg/kg on alternating days) to rats for 3 months and showed that not only were the levels of hippocampal NAD^+^ lowered by the induction of diabetes (via streptozotocin), but they were increased by the NMN treatment, which also prevented neuronal loss and memory impairment [34]. Similarly, Lee and Yang (2019) found that NR (oral gavage, 400 mg/kg/day) for six weeks improved spatial recognition memory, locomotor activity, and nest construction in mice that had had diabetes induced by a high-fat diet and streptozotocin. Additionally, Wang et al. (2020) showed that NAD^+^ precursor supplementation could be an effective neuroprotective treatment in the event of an acute episode of severe hypoglycaemia [35]. They treated rats with NMN (intraperitoneal injection, 500 mg/kg) 30 min following glucose administration for insulin-induced severe hypoglycaemia; this improved neuronal survival and attenuated cognitive impairment 6 weeks after the injury.

Although lacking evidence from clinical trials, these findings support the conclusion that NAD^+^ precursor supplementation could improve the long-term cognitive health of people with diabetes. It would be interesting if future work addressed whether routine use for the general maintenance of cognitive health by people with diabetes could also be neuroprotective in the event of an episode of severe hypoglycaemia.

## 4. Stroke

Most studies considered so far in this review have addressed whether supplementation with NAD^+^ precursors could be effective for preventing cognitive decline resulting from age-related chronic diseases. However, acute events can also be a threat to cognitive health, with long-term impairments frequently resulting from the neural damage caused by stroke. The direct administration of NAD^+^ (intraperitoneal injection, 50 mg/kg) immediately after reperfusion decreased infarct size and oedema formation and reduced neurological impairment at 48 h in a mouse model of ischemia [10]. A similar finding was made when NAD^+^ was administered intranasally (10 mg/kg) 2 h after ischemic onset in a rat model of transient focal ischemia, where it decreased infarct size at 24 and 72 h and reduced neurological deficits [36]. The same study also investigated the NAD^+^ precursor NAM but did not find any effect, and it was also observed to be ineffective at reducing infarct volume unless combined with ketamine treatment [37]. However, an additional two studies did find neuroprotective effects for NAM when administered in response to a stroke model, including improved remyelination, decreased functional deficits (intraperitoneal injection, 200 mg/kg/day) [38], and reduced infarct size (intraperitoneal injection, 200 mg/kg once) [39]. An additional study on NAM (intraperitoneal injection, 500 mg/kg) investigated its neuroprotective effects when administered before ischemic reperfusion (10–12 h or 20 min) [40]. Although delaying reperfusion is not a clinically feasible scenario, it does give some indication that existing supplementation with an NAD^+^ precursor could be neuroprotective (brain NAD^+^ and ATP levels, which otherwise dropped, were partially preserved during reperfusion) for patients in the event of a stroke. As the risk profile for dementia, diabetes, and stroke broadly overlap, if NAD^+^ supplementation was adopted for the former two conditions, this coincidence could become common.

Almost unmentioned in this review has been the NAD^+^ precursor niacin. This relative dearth of research has likely resulted from niacin’s undesirable profile of side-effects (e.g., flushing, itching, dizziness), despite it being well-characterised due to its longstanding use for the treatment of pellagra and dyslipidaemia [41]. However, some rodent studies have suggested that niacin could be useful for preserving cognitive health following stroke, with improvements reported for infarct size [9,42] and several measures of cognitive–behavioural outcomes, including the corner test, latency to move one body length, sticky-tape removal test [9], functional recovery, foot-fault scores, global test [42], and modified neurological severity scores [42,43].

## 5. Traumatic Brain Injury

Traumatic brain injury (TBI) is another acute effect with the potential to cause a loss of cognitive function. Although sports, military service, and driving accidents result in a large number of TBIs amongst younger people, the majority of hospitalisations and deaths are from older adults [44,45], principally as a result of falls. Although improved clinical management has led to increased survival, delayed neuronal death and the permanent loss of cognitive function remains a common feature [46,47].

One study investigated the effect of the intranasal delivery of NAD^+^ (20 mg/kg) following weight drop TBI in rats [46]. The authors found that neurons in CA1, CA3, and the dentate gyrus of the hippocampus were protected by the treatment, although cortical neurons were not. Delayed microglial activation was also reduced, although no measures of actual cognition were applied. Other work has focused on the potential of NAD^+^ precursor supplementation—specifically NAM—to preserve cognitive function in the event of TBI.

In a model of bilateral frontal controlled cortical impact (CCI), rats that received a 150 mg/kg/day NAM pump 30 min after TBI for seven days had smaller lesions as well as lower sensory, motor, and cognitive behavioural deficits than those that received no NAM—although working memory was not improved [48]. For unilateral CCI, a 50 mg/kg/day NAM pump 30 min after TBI for seven days reduced the initial magnitude of the injury deficit, limb use asymmetries, and the extent of cortical damage [49]. The bolus administration of 500 mg/kg of NAM following CCI improved beam walk, locomotor placing, the retention of memory, and working memory [50]. Another a study investigated the effective time course for NAM administration (50 mg/kg at 15 min, 4 h, and 8 h after CCI with boosters at five 24 h intervals) [51] and found that the tactile removal and vibrissae forelimb placing tests were improved for all time points, but that the acquisition of reference and working memory were only improved when NAM was received at 15 min or 4 h. As such, the authors concluded that the effective interval for NAM is task-dependent, allowing up to 8 h for sensorimotor ability but only 4 h for cognitive recovery. The same group also compared doses—500 and 50 mg/kg intraperitoneally after fluid percussion injury—and found that both improved the vibrissae forelimb placing test and the bilateral tactile adhesive removal test scores. However, with respect to cognition, only 500 mg/kg improved the performance of working memory [52], and neither treatment improved reference memory.

Not all preclinical trials have produced positive results, however. One study of middle-aged rats with unilateral CCI found that 50 mg/mL had no effect, and some measures of recovery were even made worse by 500 mg/mL [53]. The negative effects were speculated to be due to increased apoptosis and/or sirtuin inhibition caused by NAM; however, no investigation was undertaken. Additionally, in 2016, a multicentre, preclinical drug and biomarker screening consortia for TBI—operation brain trauma therapy (OBTT) —undertook a targeted study to determine whether NAM was a promising enough preclinical factor to warrant translation [54]. Rats were exposed to either moderate fluid percussion injury (FPI), CCI, or penetrating ballistic-like brain injury (PBBI). NAM at 50 or 500 mg/kg was then delivered intravenously at 15 min and 24 h after injury. The researchers found an intermediate effect for 500 mg/kg on balance beam performance for rats affected by CCI, but no effects were seen for any other motor tasks across any of the other TBI models. Working memory had an intermediate benefit for 500 mg/kg in the FPI model, but a negative effect for 50 mg/kg on cognitive outcome for CCI and no improvement for the PBBI model. Although the authors emphasised the potential of other treatment regimens—with particular reference to continuous infusion pumps such as those used in Goffus et al. (2010) and Vonder Haar et al.—their finding of inconsistent, intermediate-sized effects seems to have largely ended interest in NAM as an intervention for preserving cognitive health in TBI, as no subsequent works in the area were found by this review.

## 6. Healthy

Several studies have investigated the cognitive effects of NAD^+^ precursors in the absence of any specific disease of interest. This includes one of the few human trials, wherein eight men were given 30 mg of NADH by tablet and/or a placebo for four weeks in a double-blind crossover trial with a 14-week wash-out period [55]. No effect was found for ability to concentrate or feelings of fatigue. VO_2_ max and anerobic running time were also not affected, although counter-movement jump was improved. A rat model found that inducing niacin deficiency resulted in significant improvements in spatial learning, which were then lost on treatment with niacin [56]. The authors additionally compared healthy controls to NAM-treated rats and found that the NAM treatment reduced spatial learning. The authors noted that their findings should not be over-credited, particularly as niacin deficiency in humans results in the disease pellagra, which is accompanied by significant cognitive impairment. An additional human pilot study investigated the intravenous infusion of 750 mg NAD^+^ and did not observe any adverse effects, cognitive or otherwise [57].

## 7. Other Contexts

Several other health contexts have been investigated for NAD^+^ precursor supplementation’s ability to prevent cognitive impairment in individual studies. Cognitive impairment resulting from chemotherapy is an adverse effect with increasing recognition and potentially long-term impacts [58]. In cisplatin-treated mice, however, 250 mg/kg/day of NMN injected four hours before cisplatin administration for 3–4 cycles (five days on, five days off) prevented abnormal neural progenitor proliferation, neuronal morphogenesis, and cognitive function without any impact on tumour growth or the anti-tumour efficacy of the cisplatin treatment [59].

Myalgic encephalomyelitis (also termed chronic fatigue syndrome) is a complex, multisystem illness whose aetiology has not been fully defined. Fatigue, which is characterised by feelings of tiredness and a lack of energy, is well-known to negatively influence cognitive function [60]. Co-supplementation with NADH (20 mg) and co-enzyme Q10 (200 mg) daily over 12 weeks in a randomized, placebo-controlled, double-blind trial resulted in reduced cognitive fatigue perception and improved scores for FIS-40 (a fatigue impact scale with a cognitive domain) and SF-36 (a health-related quality of life scale) [61]. The individual effect of NADH was not investigated. Studies involving NAD+ precursors for the protection of cognitive health in the context of other potentially impairing conditions (e.g., epilepsy, obesity) were not found.

## 8. Conclusions

A large body of preclinical research supports the potential effectiveness of NAD^+^ precursor supplementation for preserving cognitive health across a variety of disease contexts, with the strongest evidence presented for Alzheimer’s disease and other forms of dementia. Findings have not been unanimous, however, and some investigations have reported negative effects. Additionally, human translation, where attempted, has often been unsuccessful. The major findings are summarised in Table 1. A review of the clinical trials register, clinicaltrials.gov, indicates that several trials that will investigate NAD^+^ and its precursors for cognitive protection have been initiated or have begun recruiting. These include a double-blind placebo-controlled study “NAD Therapy for Improving Memory and Brain Blood Flow in Older Adults With Mild Cognitive Impairment”, investigating niagen (nicotinamide riboside); a single-group, open-label study “Effects of Nicotinamide Riboside on Bioenergetics and Oxidative Stress in Mild Cognitive Impairment/Alzheimer’s Dementia”; a placebo-controlled crossover trial for niagen “Crossover Trial for Nicotinamide Riboside in Subjective Cognitive Decline and Mild Cognitive Impairment”; a double-blind randomised study on niagen “NAD Therapy for Improving Memory and Brain Blood Flow in Older Adults With Mild Cognitive Impairment”; and a double-blind andomized trial of nicotinamide “Nicotinamide as an Early Alzheimer’s Disease Treatment (NEAT)”. Only one completed trail was found, “The Effects of Nicotinamide Adenine Dinucleotide (NAD) on Brain Function and Cognition (NAD)” [62], for which the associated results had not been posted. However, a conference presentation which appears to have arisen from this study reported that NR was well-tolerated and increased NAD^+^, but that although fMRI (assessing blood flow) and physical function were improved, no significant difference in cognition was found. NAD^+^ precursor supplementation has received increasing attention amongst the lay population, and, with some regional variation, access to it is not generally restricted. As such, further properly controlled clinical trials are needed to provide definitive answers on its safety and effectiveness when used in humans.

## Figures and Tables

**Figure 1 nutrients-14-03231-f001:**
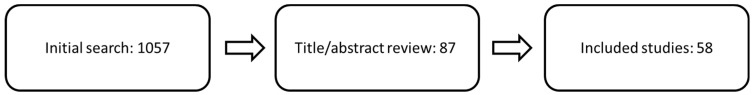
Literature review carried out in February 2022, which included the PubMed and Embase databases. Searches included terms relating to NAD+ and its precursors (e.g., NADH, NAD+, nicotinamide, NAM, nicotinamide mononucleotide, NMN, niacin, and nicotinic acid) and terms relating to cognition (e.g., cognitive, intelligence, dementia, and Alzheimer’s cognition), which were joined using Boolean operators. ‘Snowballing’ of the reference lists of the included studies was used to identify further relevant publications.

**Table 1 nutrients-14-03231-t001:** Summary of major findings.

Disease	Findings
Dementia	Preclinical research suggests NAD^+^ precursors could be effective for different forms of dementia. Head-to-head comparisons of different precursors (NR, NMN, NAM, niacin) are lacking. Work is needed to investigate the safety and effectiveness of prophylactic supplementation with NAD^+^ precursors in people at risk of age-related cognitive decline.
Diabetes	Animal models support the conclusion that NAD^+^ precursor supplementation could improve the long-term cognitive health of people with diabetes, although clinical trials are needed. Investigation is needed on the impact of routine supplementation with NAD^+^ precursors on cognitive impairment resulting from an event of severe hypoglycaemia.
Stroke	Animal studies support the conclusion that treatment with NAD+ precursors could improve cognitive recovery after stroke. The effect of supplementation prior to ischemic injury has not been investigated.
Traumatic brain injury	Almost all work on TBI has focused on supplementation with NAM. Effects have been inconsistent, with some studies reporting negative impacts on cognitive health. This field of inquiry now appears inactive.

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
