# Peer review of "Supplementation with NAD+ and Its Precursors to Prevent Cognitive Decline across Disease Contexts"

_nutrients, 2022, doi:10.3390/nu14153231_

Round 1

Reviewer 1 Report

Campbell Nutrients 2022

The author presented a literature summary of animal and clinical studies using NAD+ precursors in promoting brain health.

The topic is relevant and there are studies within this field.

However, the manuscript:

- is very descriptive

- lacks Figures and Tables

- does not go beyond listing available literature in the field. It lacks perspective and reflection. It should include benefits and drawbacks or limitations of these precursors and the studies conducted.

- mentions ‘cognitive decline’ in the title, but also includes conditions as diabetes and healthy population, so either the title or the text should be adapted

Author Response

The author presented a literature summary of animal and clinical studies using NAD+ precursors in promoting brain health.

The topic is relevant and there are studies within this field.

However, the manuscript:

- is very descriptive

It is a goal of the manuscript to provide readers with an overview of the current state of the field. The ‘Conclusions’ section has now been expanded.

- lacks Figures and Tables

Now included.

- does not go beyond listing available literature in the field. It lacks perspective and reflection. It should include benefits and drawbacks or limitations of these precursors and the studies conducted.

The major conclusion of the review is that although strong positive findings have been made in animal studies, the field is hampered by a lack of human investigation. In the absence of translational studies none of the precursors can be considered to have been thoroughly investigated for meaningful estimates to be made as to their relative strengths for actual application. Particularly as head-to-head trials, even in animals, are lacking. The evidence that exists has been presented and evaluated. I do not believe that I would be advantaging readers by trying to go beyond it.

- mentions ‘cognitive decline’ in the title, but also includes conditions as diabetes and healthy population, so either the title or the text should be adapted

I do not fully understand the issue with this point. Cognitive decline is a symptom not definitely attached to any specific disorder. Diabetes in particular is well known to be associated with premature cognitive decline. I do not understand how this context makes the title inappropriate.

Reviewer 2 Report

The manuscript summarizes the research findings on the preventative effect(s) of NAD supplementation on cognitive impairment/decline across various diseases, with a focus on primary measures of cognitive function and behavior. The author concludes that most of the relevant studies were conducted based on animal models with majority of the findings being positive despite some of the studies reporting insignificant or adverse effect. The author further points out that well controlled clinical studies are needed to provide definitive answers to the potential impact of NAD supplementation on human cognitive functions impaired by various diseases. The manuscript is nicely written and provides specific information such as dose and frequency as well as duration. It also highlights promising avenues and knowledge gaps. All of these can be very useful.

The manuscript can benefit from the following improvements:

1)      Add a research strategy scheme to include how and which database(s) the author gather the included information.

2)      Add a table to summarize the major findings.

3)      Provide current clinical trial status on the topic of NAD supplementation on cognitive impairment/decline.

4)      Clarify what “the 2021 systematic review” refers to in “Several additional studies with relevant findings were not included in the 2021 systematic review”.

5)      Clarify why obesity and epilepsy-associated cognitive deficit are left out of the picture in this manuscript.

Author Response

The manuscript summarizes the research findings on the preventative effect(s) of NAD supplementation on cognitive impairment/decline across various diseases, with a focus on primary measures of cognitive function and behavior. The author concludes that most of the relevant studies were conducted based on animal models with majority of the findings being positive despite some of the studies reporting insignificant or adverse effect. The author further points out that well controlled clinical studies are needed to provide definitive answers to the potential impact of NAD supplementation on human cognitive functions impaired by various diseases. The manuscript is nicely written and provides specific information such as dose and frequency as well as duration. It also highlights promising avenues and knowledge gaps. All of these can be very useful.

The manuscript can benefit from the following improvements:

1)      Add a research strategy scheme to include how and which database(s) the author gather the included information.

The research strategy as well as databases has been added as figure 1.

2)      Add a table to summarize the major findings.

The requested table has been added to the manuscript and is referenced in the text.

3)      Provide current clinical trial status on the topic of NAD supplementation on cognitive impairment/decline.

A new section has been added to the Conclusions addressing this issue as follows:

“A review of the clinical trials register; clinicaltrials.gov, indicates that several trials have been initiated or begun recruiting to investigate NAD+ and its precursors for cognitive protection. These include a double blind placebo controlled study "NAD Therapy for Improving Memory and Brain Blood Flow in Older Adults With Mild Cognitive Impairment", investigating Niagen (nicotinamide riboside), a single group, open label study "Effects of Nicotinamide Riboside on Bioenergetics and Oxidative Stress in Mild Cognitive Impairment/Alzheimer's Dementia", a placebo controlled crossover trial for Niagen "Crossover Trial for Nicotinamide Riboside in Subjective Cognitive Decline and Mild Cognitive Impairment", a double blind randomised study on Niagen "NAD Therapy for Improving Memory and Brain Blood Flow in Older Adults With Mild Cognitive Impairment", and a double blind randomised trial of nicotinamide "Nicotinamide as an Early Alzheimer's Disease Treatment (NEAT)". Only one completed trail was found, "The Effects of Nicotinamide Adenine Dinucleotide (NAD) on Brain Function and Cognition (NAD)" (Orr, Kotkowski et al. 2020) which did not have associated results posted. However, a conference presentation which appears to have arisen from the study reported that NR was well tolerated and increased NAD+, but that although fMRI (assessing blood flow) and physical function were improved, no significant difference in cognition was found.”

4)      Clarify what “the 2021 systematic review” refers to in “Several additional studies with relevant findings were not included in the 2021 systematic review”.

That refers to the findings of reference 16 which are introduced further up in the text. To better structure results I have repositioned this paragraph to the one in which the systematic review is introduced and initially discussed. I have also referenced the systematic review in the text.

5)      Clarify why obesity and epilepsy-associated cognitive deficit are left out of the picture in this manuscript.

Unfortunately no studies were identified in my review of the literature where the impact of NAD+ precursors the effect of NAD+ precursor supplementation on cognitive health was studied in people affected by obesity or epilepsy.

The following text has been added: 

"Studies where NAD+ precursors for the protection of cognitive health from other potentially impairing conditions (e.g. epilepsy, obesity) were not found."